# Leveraging History to Predict Infrequent Abnormal Transfers in Distributed Workflows [note 1]

**DOI:** 10.3390/s23125485

**Published:** 2023-06-10

**Authors:** Robin Shao, Alex Sim, Kesheng Wu, Jinoh Kim

**Affiliations:** 1EECS, University of California at Berkeley, Berkeley, CA 94720, USA; robin_shao@berkeley.edu; 2Lawrence Berkeley National Laboratory, Berkeley, CA 94720, USA; asim@lbl.gov (A.S.); kwu@lbl.gov (K.W.); 3Computer Science Department, Texas A&M University, Commerce, TX 75428, USA

**Keywords:** network transfer, slow connection, prediction, machine learning, scientific computing

## Abstract

Scientific computing heavily relies on data shared by the community, especially in distributed data-intensive applications. This research focuses on predicting slow connections that create bottlenecks in distributed workflows. In this study, we analyze network traffic logs collected between January 2021 and August 2022 at the National Energy Research Scientific Computing Center (NERSC). Based on the observed patterns, we define a set of features primarily based on history for identifying low-performing data transfers. Typically, there are far fewer slow connections on well-maintained networks, which creates difficulty in learning to identify these abnormally slow connections from the normal ones. We devise several stratified sampling techniques to address the class-imbalance challenge and study how they affect the machine learning approaches. Our tests show that a relatively simple technique that undersamples the normal cases to balance the number of samples in two classes (normal and slow) is very effective for model training. This model predicts slow connections with an F1 score of 0.926.

## 1. Introduction

Scientific applications for climate modeling [1], bioinformatics [2], particle physics [3], and so on often require a large amount of data from geographically dispersed sites. For instance, a Large Hadron Collider (LHC) experiment produces petabyte-scale data and distributes them to 160 computing facilities around the world [4]. There are thousands of physicists making use of some portions of this data collection to conduct their research. Such distributed scientific workflows rely heavily on the networking infrastructure for moving their data. Scientists could usually receive their data files promptly with the currently deployed software and hardware [3,4,5,6,7]. However, occasionally, some of these data movements are much slower than expected. This work aims to investigate whether such slow data transfers could be predicted before the start of the data request. Having such information would allow the data management system to make alternative arrangements and improve the overall effectiveness of the data infrastructure.

There has been a considerable amount of work on monitoring and analyzing network performance [8,9,10]; however, much less attention has been given to the understanding and prediction of low-performing communications. However, such slow data transfers could easily become the bottlenecks of a large distributed workflow [4,11,12]. In this study, we explore various properties of slow connections using a set of network traffic logs from a scientific computing facility (NERSC (https://www.nersc.gov/ (accessed on 5 June 2023))) and develop a prediction mechanism to identify unexpectedly low-performing data transfers.

We start with nearly two-year’s worth of network traffic monitoring data produced by tstat [13]. The traffic logs are from dedicated data transfer nodes (DTNs) for moving the large amounts of data needed for mission-critical scientific applications [14]. This study expands on the workshop report [12] as follows: (1) The previous work only used data records from one of the DTNs at NERSC (dtn01) and only covered five months, from January–May 2021, while this study is based on an extensive dataset collected from all four public DTNs covering nearly two-year’s time (from January 2021 to August 2022). (2) This work systematically addresses the issue of far fewer slow transfers than normal ones by defining a set of sampling strategies for mitigating the class-imbalance concern. (3) This work also explores advanced ways of defining what should be considered slow transfers, which includes a non-static threshold defined as a function of transfer size, to consider the correlation between the size and network throughput.

By exploring features from the network monitoring data, we are able to devise effective decision tree-based classification techniques to identify slow transfers before the start of the transfer. Overall, the key contributions of this work include the following:We present a set of stratified sampling techniques to address the challenge introduced by the fact that are far fewer slow transfers than normal ones. The best sampling approach allows us to achieve an F1 score of 0.926 for predicting under-performing transfers.We devise a strategy to capture the network state by utilizing information from recently completed file transfers. (Even though this was a key technique used previously in [12], we feel it is still worthwhile to draw attention to it because it is an effective approach that could be effectively used in many application scenarios.) We define a set of features engineered from the most recent transfer from the same subnet as the transfers from a similar location, and the host may show similar behaviors.This study utilizes extensive network monitoring data from an active scientific computing facility. This larger data collection allows us to explore different sampling strategies and definitions of slow traffic and show the relative importance of the individual features.

This paper is organized as follows. Section 2 describes this study’s data (tstat) and introduces our exploratory data analysis that provides an understanding of the nature of data transfers in scientific facilities. In Section 3, we present our prediction methodology with feature engineering, learning algorithms employed, and sampling strategies used for optimizing models, and we report the experimental results and observations with the experimental setting in Section 4. Finally, a summary of the previous work closely related to our study is provided in Section 5, and we conclude our presentation with future directions in Section 6.

## 2. Data Description and Exploration

This section provides key observations from our exploratory data analysis performed to better understand the characteristics of data transfers measured in the tstat format.

### 2.1. Description of tstat Data

Simply, tstat (http://tstat.tlc.polito.it/measure.shtml (accessed on 5 June 2023)) is a logging tool for network traffic flows. This tool is deployed on NERSC data transfer nodes (DTN). Since the DTNs are dedicated to moving data files among the different computing facilities, each of the tstat flows corresponds to a file transfer.

Viewed as a data table, the tstat dataset has 116 columns, each corresponding to some feature about the TCP flow, and each row is about a network flow. The features recorded include communication activities with the address information, such as source IP address, destination IP address, the number of bytes transmitted in payload, the number of retransmissions, start time, flow duration, minimum/maximum/average round trip time (RTT), etc. This is the basic information we would use for our predictions. However, it is essential to note that most of these values should only be available after the start of the transfer. In fact, we only know about the IP addresses and the number of bytes in the file in the prediction time. With limited known information, a significant challenge of this study is to define new features to predict network performance for individual transfers.

When analyzing the tstat data, the second challenge is that what we assume as unusually slow is a tiny fraction of the total number of observed network flows. This challenge is common to many two-class classification problems with a significant difference between the majority class and the minority class in the notion of their quantity [15,16]. Machine learning models generally have a hard time learning about the minority class when a significant imbalance exists in the dataset.

We perform a train–test split to ensure that our model can generalize well to new data. For this study, we use all the data collected in 2021 as the training set and all the data from 2022 as the testing set. Note that the tstat data used for our study is from the operational log of a large computing facility, which contains scientific data transfer measurements over high-speed backbone communication links. Given the large number of data transfers recorded (as shown in Table 1), we use subsets of data records from training and testing data to keep the time needed for training and testing reasonably modest.

### 2.2. High-Level Observations about Data Transfers

In our previous study [12], we only had access to tstat records during the first five months of 2021 on DTN01. For this study, we have access to a much larger dataset covering a period from January 2021 through August 2022. Furthermore, the data collected include tstat logs on all four DTNs. Therefore, the data collection used for this study not only has many more data records but also has a greater variety of file transfers.

For scientific workflows, the relatively large file transfers experiencing low network performance create long delays. Thus, our work focuses on network flows that are relatively large in size. Specifically, we consider the performance of file transfers, where the file sizes are larger than 1 × 10^6^ bytes (denoted as 1 MB), which is also helpful for eliminating control channels used for exchanging file exchange commands. Figure 1a shows the distribution of transfer size in a log scale, while Figure 1b provides the distribution of throughput. Table 1 has the counts of these large transfers vs. the total number of transfers. From these total counts, we see that the total number of transfers is within a factor of two among four DTNs, while the number of large transfers differs considerably.

In detail, DTN02 carries about 40 million large transfers, while the total number of large transfers by all four DTNs is less than 41 million. That is, DTN02 carried about 97.6% of the large transfers. This is because many large physics projects have set up automated data management tools to use this particular DTN. Because these automated data transfers are between large computing facilities that carefully monitor their storage and network performance, these file transfers also enjoy good transfer throughput as shown in the next section.

In our previous study with data transfer involving DTN01, we found most of the slow transfers are from IP addresses that are infrequently used, often appearing only once or a small number of times [12]. These occasional uses might involve a personal laptop in a work-from-home scenario or a user in an internet cafe. It is unlikely that such use cases would become the dominant mode of operation for large scientific collaborations. Instead, we focus on eight Class C network addresses that transfer data to NERSC most frequently, such as those involved in the automated transfers at DNT02. These eight Class C IPv4 networks are from four institutions, ‘Imperial College’, ‘SLAC’, ‘Fermi Lab’, and ‘CERN’, from three countries, England, Switzerland, and United States. (The eight Class C IPv4 addresses are 146.179.234, 146.179.232, 146.179.233, 134.79.138, 131.225.69, 128.142.209, 128.142.33, and 128.142.52, where the first three are from Imperial College and the last three are from CERN.) Note that these frequently used sites are well managed, and thus, the data transfers suffer from low performance less frequently, which increases the imbalance between normal and slow transfers, which signals the necessity of tackling the class-imbalance concern in our prediction study.

Another observation from Table 1 is that the number of records to be studied is quite large, which imposes high computational costs in the analysis process. We keep the computation cost for learning and testing down by performing these tasks on a sample of the training and testing data.

### 2.3. Data Cleaning and Statistics about tstat Data

To prepare the data for our prediction effort, we first filter the data records to keep only those with file sizes larger than 10^6^ bytes (1 MB). Additionally, we filter by the minimum round trip time (RTT) and only keep those with a minimum RTT greater than 1 ms in the data clean-up process based on the assumption that those transfers are local communications or inadequately recorded.

We then extract several crucial features not present in the recording produced by tstat. The first feature extracted is the transfer throughput computed as the ratio of transfer size and transfer duration. By convention, we report this as bits per second (bps). Another useful feature is the country code. Since we analyze the tstat data measured at NERSC, one end of every data transfer is always NERSC. We look up the country where the other end of the transfer is. This is done through a lookup on the GeoIP2 database.

Intuitively, the size of the transfer would impact the data access performance. Figure 2a shows how throughput distributes across different file sizes. Broadly, the throughput appears to be positively correlated with the file size. In the figure, we colored the data points by countries, where only the top three countries are shown to avoid cluttering. We can see that the fastest transfers (≥1 Gbps) are primarily from the United States, which is intuitive because of the closer physical distance (to NERSC) and higher bandwidth offered by the Energy Science Network (ESnet (https://www.es.net/ (accessed on 5 June 2023))) backbone. Figure 2b closely focuses on slow transfers with less than $10^{6}$ bps throughput. In this case, we see that the slowest transfers are nearly six orders of magnitude slower than the hardware limit (>10 Gbps). Therefore, it is worthwhile to study these slow transfers and to find alternative options to avoid such extremely poor performance.

Another important feature that affects data transfer throughput is round-trip time (RTT). Figure 3a shows a scatter plot of throughput again the minimum RTT. This plot shows distinctive vertical stripes due to many transfers from the same computer sites (with the same minimum RTT) but having very different throughput ranges. Within the United States, there are two clear stripes, one around 10 ms RTT, which is within the San Francisco Bay Area, and the other around 60 ms, which is the RTT for communicating with sites on the East Coast of the United States. The minimum RTT from NERSC to European countries is between 130 and 200 ms. Overall, we expect higher RTT to lead to lower throughput, which is true; however, many other factors could impact the actual throughput, which explains the wide throughput range. The tstat measurement records four features related to RTT, minimum, maximum, mean, and standard deviation. Among these four, our observation shows that the minimum RTT has the highest correlation with throughput, with a coefficient of −0.18.

When a network packet is determined to be lost, the data transfer system will retransmit the packet. A high retransmission rate indicates that the networking system is not functioning correctly. We define the retransmission rate as the ratio between retransmitted bytes and the total bytes transferred in the tstat record. Our data collection shows no clear correction between the throughput and retransmission rate when the retransmission rate is less than 0.01. Figure 3b shows a scatter plot of throughput against the retransmission rate, where we can see a visible trend. The correlation coefficient between the throughput and retransmission rate in the log–log scale is −0.412. The trend line shown in the figure suggests that, on average, the throughput is proportional to ${\mathrm{retx}}^{-0.52}$.

The details of individual features defined in this study are described in Section 3.3. After data cleaning and filtering based on different features, we obtained a dataset size of over 24 million instances, which are transfers from major transfer sites with relatively large transfer sizes and round-trip times.

## 3. Prediction Methodology

Once a data transfer is completed, we can compute the throughput, which tells us whether the transfer *was* slow or not. We aim to make this prediction at the start of the transfer. If we could make this prediction reliably, we could use the prediction to make alternative arranges when a data transfer *is expected* to be slow. As indicated earlier, however, we face at least two challenges in this prediction task. There is a significant class imbalance since slow transfers are rare events. In this work, we explore several different stratified sampling techniques to address this class-imbalance problem. The second challenge is the lack of information at the start of the file transfer. To address this challenge, we look into the recently completed file transfers from the same site.

### 3.1. Defining “Slow” Transfers

In our work, we need to come up with a definition of “slow” transfers. We define slow transfers based on the throughput information (bps). A simple choice might be to define a firm threshold. For example, we can declare all transfers whose throughput is less than 1 × 10^6^ bps as slow and all other transfers as non-slow (“normal”). This is the choice used in our initial work [12]. We will continue to use this choice but also attempt to explore alternative options as described next. There are about 9000 transfers slower than $10^{6}$ bps, which is 0.037% of all transfers used in modeling.

In addition to the static threshold (e.g., 1 Mbps), we consider an alternative threshold defined as a function of file size. In fact, TCP throughput has a high correlation with the file size transferred [17]. From Figure 2a, we see a clear positive correlation between the throughput and file size. In later tests, we observe a wide gap between normal transfers and slow transfers (after stratified sampling), which suggests that we might be able to draw a power–law line in Figure 2a to better separate“slow” and “normal” transfers. From our empirical study, we observe that the following boundary line appears to be the simplest (where tput is throughput):(1)$$tput=10^{3}\times size^{1/2}$$

### 3.2. Sampling Strategies

Given fewer than four slow transfers out of one thousand non-slow ones, it is hard for learning methods to extract a model for the slow transfers [15,16]. One classic strategy to deal with the class-imbalance issue is the reliance on stratified sampling that produces a balanced dataset among different classes. In our case, we need to identify two classes of events: slow vs. normal (non-slow). A straightforward approach would be to define two strata, one for slow transfers and the other for normal transfers. Within each stratum, we may choose a different sampling method. Since there are a large number of normal transfers, we may further divide them into more strata, for example, dividing them into a set of bins across the throughput space.

Additionally, applying a sampling technique is beneficial for managing the computational cost. To make training complete in a reasonable amount of computing time, we use a fraction of the training data (from 2021) and the testing data (from 2022). We experimented with a number of different sampling techniques. We next introduce four of them that are representative of different considerations (also summarized in Table 2):

*Baseline* (train1/test1): The baseline method is a uniform random sampling of the 41 million transfers with large files. Following this sampling method, we sampled 10,000 transfers from each training and testing period and named the two subsets as train1 and test1. Figure 4a shows a histogram of train1 subset. Note that this subsample does not address the class-imbalance problem. In particular, the test1 contains only three instances of slow transfers with throughput less than $10^{6}$ bps.*Stratified 2* (train2/test2): To address the class-imbalance problem, all of our stratified samplings keep all slow transfers and select different samples from the normal transfers. Since there are 8986 slow transfers, the total number of normal transfers selected is also 8986. The simplest method to select the normal transfers is to sample them uniformly. This approach of selecting data records from the training data (from the year 2021) is named train2, and the similarly selected subset from the year 2022 is named test2. The distribution of train2 is shown in Figure 4b. This figure shows a clear gap between $10^{6}$ and $10^{7}$ bps. Training with this dataset might not be able to learn that the actual decision boundary is at $10^{6}$ bps. On the other hand, test results on test2 might be very good since there are fewer data samples near the decision boundary to challenge the classifier.*Stratified 3* (train3/test3): To put more data samples near the decision boundary of $10^{6}$ bps, we employ another stratified sampling on the normal transfers. Specifically, we divide the normal transfers into bins based on the logarithm of their throughput. This binning choice was selected after experimenting with a number of different approaches. Since the concern with train2 is that there might not be a sufficient number of samples near the decision boundary, our choice here is to place more samples near the decision boundary. To choose from the normal transfers, we select a number of samples from a bin that is inversely proportional to its lower bin boundary, which samples significantly more normal transfers with lower throughput than those with higher throughput. The subset of data thus selected from the training data (from the year 2021) is named train3; similarly, we also created test3 from the testing data from the year 2022. The histogram of the train3 is shown in Figure 4c. With this distribution, we anticipate that the training task will lead to a more precise model because there are more points near the decision boundary. This fact would also make testing on test3 have lower performance because the test case near the decision boundary would be challenging for the classifier.*Stratified 4* (train4): Another stratified sampling strategy we examine selects the same number of records from each logarithmic bin for normal transfers while keeping all slow transfers, producing a training sample named train4. The resulting distribution is shown in Figure 4d.

### 3.3. Extracting Network States from Recently Completed Transfers

Among the features tstat collects, only the transfer size and IP addresses are known at the start of a file transfer. From our earlier exploration, we found that these two features are insufficient to accurately predict the transfer throughput. To make effective predictions, we created features derived from recently completed transfers. The previous exploration of the tstat data shows that such features as file size, transfer duration, RTT, and retransmission rate highly influence the final transfer throughput. These features from the most recently completed transfer involving the same source–destination pair could be used as a proxy to represent the network state for the current transfer.

For such information, freshness would be critical for its usefulness. To make it easier to find past information for our prediction task, we relax the matching of the source and destination as long as the first three octets of the IPv4 address, i.e., the two IP addresses, are in the same Class C network. For example, the four SNDs have IPv4 addresses in the same Class C network. Thus, they are regarded as equivalent to locating a recently completed transfer. Similarly, if we are looking at recently completed transfers for a remote host with IPv4 address A.B.C.D1, we look for all completed transfers from remote hosts whose IPv4 address starts with ABC and then select the one that was completed most recently. Among the large scientific data centers, their DTNs generally are on the same Class C network and have typically uniform hardware configuration as well as the same accesses to the same storage system at the backend. Therefore, it is reasonable to regard the hosts on the same Class C network as identical (at least in scientific facilities). This assumption may not always be satisfied but it appears to provide somewhat useful information for our prediction task.

We also include the ratio between the current transfer size and the previous transfer size as a feature, which can be helpful in measuring the difference between the current and previous transfers. Additionally, we record the time difference between the recently completed transfer and the current transfer as a feature. The intuition here is that the larger the time difference, the less similar the two transfers might be. Table 3 shows a summary of all features we extract from the recently completed transfers.

Before actually using the data records for training and testing, we apply a normalization that translates all numerical values to be between 0 and 1 (for learning purposes). Additionally, tstat captures the source and destination address in IPv4. Since one side of the communication is always a NERSC DTN (because the transfer log comes from NERSC), we only keep some information about the remote host. In fact, the only information we keep about the remote host is which country the IP address is registered in the GeoIP2 database. In the final data table used for training and testing, only the three most frequently occurring countries are kept (United States, Switzerland, and the United Kingdom), which includes the eight computing sites mentioned in Section 2.2.

### 3.4. Prediction Algorithms

After setting up all the features and labels, we built binary classification models to predict the low-performance transfers. The training and testing data include features size, country, and those computed from recently completed transfers given in Table 3.

We explored several classification models, including decision trees, random forests, and extreme gradient boosting (XGBoost). Tree-based models were particularly effective because we have a relatively small number of features in the data, and we allow the decision tree algorithm to explore all possible combinations of features by shrinking the training set size using sampling described in Section 3.2. We began with the decision tree model and found that it performed the best. Details of the evaluation are discussed in Section 4.

The random forests method combines multiple decision trees by bagging and training each tree on a different sample of the dataset. The final prediction is the majority vote of all the trees. We used all the features we created and grid search to find the best hyperparameters. We expected that XGBoost would outperform the decision tree model, as it is one of the most effective supervised learning methods [18]. It builds trees on the residual of the previously fitted tree. We tried XGBoost trained with all the features available, the feature combination that works best for the decision tree, and the top important features based on the try-all decision tree method. The hyperparameter choice for XGBoost is based on the best-performing XGBoost models from our previous study. We did not try further hyperparameter options, as the performance is not close to that of the decision tree models. We observed that our approach with a decision tree with a well-crafted set of chosen features outperforms the ensemble-based methods (random forests and XGBoost) as discussed in the next section.

## 4. Evaluation

In this section, we share the evaluation results with the experimental setting and our observations and findings made from the extensive experiments.

### 4.1. Experimental Setting

To build our classification model, we used a dataset from all DTNs, which consists of over 40 million network streams (file transfers with file size > 1 MB) and 10 features. Recall that we used transfers in 2021 for training and transfers in 2022 for testing (more in Section 2.1).

To measure the prediction performance, we basically refer to the conventional confusion matrix for binary classification, consisting of TP (true positive), FP (false positive), FN (false negative), and TN (true negative). Intuitively, the fraction of slow connections is small, while the majority of connections would perform normally. Hence, reporting the simple accuracy measure may misguide the audience. We measure the prediction performance using the *F1 score*, a harmonic mean of *precision* = $\frac{TP}{TP+FP}$ and *recall* = $\frac{TP}{TP+FN}$. The metric of the F1 score is defined as $F1\;\mathrm{score}=2\times \frac{Precision\times Recall}{Precision+Recall}$. A greater F1 score indicates better performance in prediction.

### 4.2. Best Model Performance

To address the class-imbalance problem in our attempt to predict slow file transfers, we designed four different stratified sampling techniques in Section 3.2. To evaluate their effectiveness, we trained with all combinations of training and testing sets shown in Table 2. We primarily utilized the try-all decision tree method, which previously demonstrated the best performance during the first phase of our research [12].

In Table 4, we present the results sorted by F1 score, the primary performance metric used throughout our study. Even though it is not meaningful to compare the F1 scores of different testing sets directly, the results from Table 4 suggest that test2 might be a better test set than test3, which is, in turn, better than test1. Overall, the best performance training and testing combination achieved an F1 score of 0.926, which is a considerably higher value than the naive combination of train1-test1. This indicates that the stratified samples are effective in addressing the class-imbalance problem.

The test set test1 is a uniform random sample of the whole dataset. It has low F1 scores because of a lack of slow events—recall that there are only three slow transfers out of 10,000 data records in test1, see Section 3.2. In this case, simply predicting every test case as normal would lead to a prediction accuracy of 99.97% but 0 true positive cases. Even though the precision and recall values shown in Table 4 are not 0, they are quite low. The more complex stratified sampling techniques were designed to address this class-imbalance problem and improve prediction performance.

We include test3 in our analysis because it presents a challenging test for our model, as most of the data records fall close to the classification threshold, where most misclassifications occur. With test3, all recall values are nearly 100%, while all precision values are about 50%, which suggests that the trained models are effectively declaring all testing samples to be “Positive” (i.e., slow transfers). We speculate that the normal cases in test3 are too close to the slow cases for the decision tree models to differentiate.

The four top-performing combinations in Table 4 are with test2, which suggests test2 to be more well balanced. Since test2 has the same distribution as train2 in Figure 4, we see a significant gap in the histogram between $10^{6}$ and $10^{7}$ bps. This gap allows a classifier to make mistakes, while still classifying most of the test set correctly, therefore achieving a good F1 score. By construction, test2 and train2 follow the same relatively straightforward stratified sampling, using only two strata: one for slow transfers and one for normal transfers. They both keep all records from the minority class (i.e., slow transfers) and select a matching number of samples from the majority class through random uniform sampling. This approach minimizes the changes to the distribution of the majority class. For the remainder of this study, we chose to use test2 for further analyses.

Next, we examine the top-performance training samples in more detail. The best-performing model is trained with train2, and a scatter plot of the testing results is shown in Figure 5a. Each dot in this figure represents a transfer in test2, with slow transfers in orange and normal transfers in blue. The solid green line indicates the true threshold ($10^{6}$ bps). Misclassifications occur when an orange dot is placed above the line or a blue dot is below the line. The overall prediction accuracy is high, especially for slow transfers. However, many false positives are far above the threshold indicated by the green line. Figure 5b shows two histograms using the same colors as in Figure 5a, with a density plot providing a clearer view of where the misclassifications occur. It is surprising to see the orange curve having a peak below the blue peak that is far from the decision boundary of $10^{6}$ bps.

Figure 6 shows the prediction results of the train4-test2 pair. The scatter plot (Figure 6a) shows that misclassified cases are primarily near the green line (representing $10^{6}$ bps decision boundary). Almost all normal transfers smaller than $10^{7}$ bytes are misclassified, while larger file transfers are predicted correctly. This observation is also verified by the histograms of the two predicted classes in Figure 6b. Normally, we expect the misclassification to happen near the decision boundary, and we designed train3 and train4 to have more training cases near the decision boundary. Since the test results in Figure 6 match this expectation, we say that these stratified sampling strategies behave well. Overall, we say that train4 has more training cases far away from the decision boundary than train3. The fact that training with train4 achieves a higher F1 score than training with train3 suggests that having those training cases far away from the decision boundary is important.

### 4.3. Impact of Features

As mentioned in the previous section, the testing using the most simple stratified sampling performs the best. We next conduct a feature importance study in ways similar to our previous study. We count the number of occurrences of features in the top-performing decision tree models. This time, we decide to include the top three combinations from the train–test pairs that use test2 as the testing set (shown in Table 4).

The top five features and their counts are shown in Table 5. It is unsurprising that prev_tput occurs in all 12 top models. Surprisingly, the feature size does not appear on this list, while it is shown to be the more important one based on traffic from DTN01 [12]. The size information does show up in size_ratio as the third most influential feature in Table 5. We believe this change in feature importance to be due to differences among the types of file transfers on different DTNs. Furthermore, this analysis focuses on data transfers from popular sites that are more likely to be well tuned; in the earlier study [12], we observed many slow transfers associated with large RTT values and infrequently used IP addresses, which points to uncommon workflows on not-so-well-tuned network connections.

In addition to the decision trees, we also trained our sampled data on a random forest model. Due to the relatively small number of features used, the decision tree training process can meet all possible feature combinations for the decision tree. Thus, we do not expect the random forest models to achieve higher performance. The random forest model with all features available achieves an F1 score of 0.82, which is indeed lower than the 0.926 achieved with the decision tree model. However, a random forest has a useful function of feature importance that automatically calculates the rank of individual features by the random forest model itself. The bar chart in Figure 7 displays the importance of each feature in sorted order. As we encoded the country feature using one-hot encoding, it is displayed as the United Kingdom, Switzerland, and the United States. The top features are consistent with what we obtained from the decision tree model, with prev_tput and country being the two most important features in both lists. The size-related feature and RTT-related features also have high importance.

We used the top features we derived from training an XGBoost model. Similar to the previous study, it has performance closer to the decision tree but is less efficient and accurate. It has an F1 score of 0.879.

### 4.4. Alternative Threshold Setting

Thus far, the training cases are created with a simple static threshold of $10^{6}$ bps, while the typical transfer throughput grows with the transfer size. After examining Figure 5a and Figure 6a, we propose to test a new decision boundary defined by Equation (Equation 1).

For files of 1 MB, these decision criteria still classify those transfers less than 1 Mbps as slow. However, as the file sizes grow, a slow transfer’s maximum speed would gradually increase. For a file size of 10 GB, those transfers that are slower than 100 Mbps would be classified as slow. From our data collection, this creates more slow transfer events. There are nearly 93 thousand slow transfers, 10 times more than under the 1 Mbps threshold setting. In our stratified sampling, we correspondingly increase the number of samples we take from the normal transfers. This increased training sample size might increase the effectiveness of the classification model. We created a training and testing set using basic stratified sampling with slow and normal as the only two strata. We kept all the data points from slow transfers and sampled 93 thousand transfers from normal transfers. The best F1 score we achieved with this threshold so far is 0.88, but we have yet to fully explore the best way to predict transfer performance based on this threshold.

Figure 8 shows that this threshold significantly improves our ability to control false positives, but more false negatives occur. This model may be the best choice if minimizing false positives is the main concern, but we can still improve its performance by addressing the clear patterns observed in the false negatives.

## 5. Related Work

Monitoring network traffic is one of the essential tasks in network operations and management for detecting anomalous events and estimating network performance. In scientific computing, traffic monitoring is also significantly crucial for supporting ever-increasing data-intensive scientific exploration and computing. In particular, identifying elephant flows is a critical problem, as the flows consume significant amounts of network capacity. A study in [19] introduced an algorithm estimating the traffic volume of individual flows, which is used to detect the elephant flows’ total byte count. The authors defined two hash tables recording a counter representing the volume of the flow with the associated flow ID from the packet trace, which is then used to detect elephant flows showing a pre-defined threshold. In [20], the authors tackled the problem of the classification between elephant (large transfer) flows and mice (small) flows. This previous study takes an unsupervised learning approach, and the presented clustering scheme (based on the Gaussian mixture model) produces two clusters (one for elephant flows and the other for mice flows) from the NetFlow data. While highly important to identify elephant flows, our study focuses on predicting slow connections that significantly impact data-intensive scientific applications.

There have been several studies analyzing tstat data. In [21], the authors presented a classification mechanism to detect the low throughput time intervals. The classification mechanism consists of two phases: assigning binary classification labels for each time window (anomalous or not) and performing actual classification by constructing a supervised learning model using the assigned label information. Another study in [22] evaluated deep learning models, including multilayer perceptron (MLP), convolutional neural network (CNN), gated recurrent unit (GRU), and long short-term memory (LSTM), in order to predict network performance (aggregated throughput) for each time interval. While these studies focused on analyzing tstat data based on time windows, our study focuses on connection-level prediction.

Sampling is widely considered for dealing with the concerns of class imbalance and scalable analysis in machine learning [15]. Sampling strategies may have significant impacts on the performance given the fact that not all samples are equally important [23,24]. Previous studies in [15,25,26,27,28] considered utilizing sampling strategies (including stratified sampling) for mitigating the impact of the imbalance between malicious traffic (minority) vs. normal traffic (majority) in network intrusion/anomaly detection. Sampling has also been considered in the Internet of Things (IoT) setting for class imbalance in anomaly detection [29] and data fusion, reducing data redundancy in the sensed data [30]. In this study, we investigated a set of sampling strategies for improving classification performance, including bell-shaped sampling and bin-based sampling for the problem of network performance prediction.

There are several other related areas to this study. Feature extraction and selection are crucial for applying learning schemes for prediction tasks [31]. As discussed, we primarily rely on past transfer information for extracting features. Log analysis can be performed through unsupervised techniques [32], while we simply utilize the raw data provided in a tabular format (tstat). Additionally, load balancing can be an option for facilitating data transfers [33]. For simplicity, this study does not assume a load-balancing function (e.g., transferring a file from multiple facilities). Lastly, our study is closely related to time series analysis. This study utilizes decision trees, making the prediction based on the latest transfer. Extending the history information to multiple previous transfers may need to consider deep structures [34,35,36], which would be a future investigation of this study.

## 6. Conclusions and Future Directions

This study explores tstat logs collected on data transfer nodes at NERSC. A key objective is to use such information to predict slow file transfers before the start of the operation. Our exploration of the network measurement data reveals several features correlated with transfer throughput. However, most of them are only available after the transfer. To predict the start of the transfer, we defined a set of new features based on the more recently completed transfer between the same source and destination networks. The second challenge we need to overcome is the significant imbalance between normal and abnormally slow transfers. To overcome this challenge, we devised several stratified sampling techniques. Our tests showed that one of the stratified sampling techniques could significantly outperform a naive approach without stratification. The best model trained on a stratified sample was able to achieve an F1 score of 0.926, while without stratification, the same F1 score was only 0.566. This best-performing stratified sampling consists of only two strata: one for each class considered. It keeps all records from the minority class and randomly selects the same number of cases from the majority class to create balanced training and testing sets.

For future work, we are interested in further exploring options for stratified sampling, more advanced learning techniques for model creation, and more feature engineering approaches that better use the recently completed transfers.

## Figures and Tables

**Figure 1 sensors-23-05485-f001:**
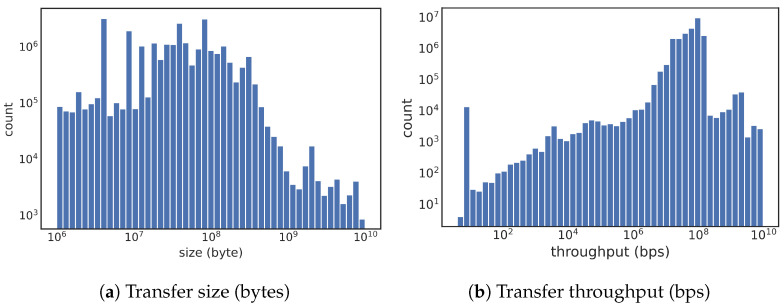
Histograms of file sizes and transfer throughput of data transfers to NERSC (size > 1 MB).

**Figure 2 sensors-23-05485-f002:**
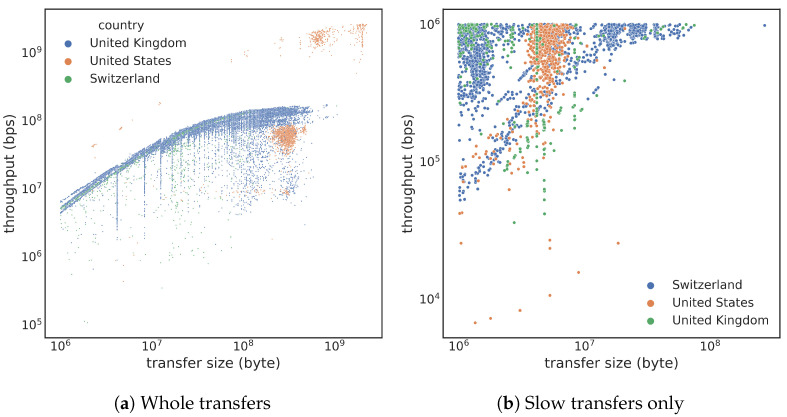
Scatter plot of throughput against transfer size (colored by country): (**a**) larger transfers typically achieve higher throughput; (**b**) slowest transfers achieve only a few kilobits per second ($10^{3}$ bps), although the networking hardware is capable of servicing greater than 10 Gbps ($10^{10}$ bps).

**Figure 3 sensors-23-05485-f003:**
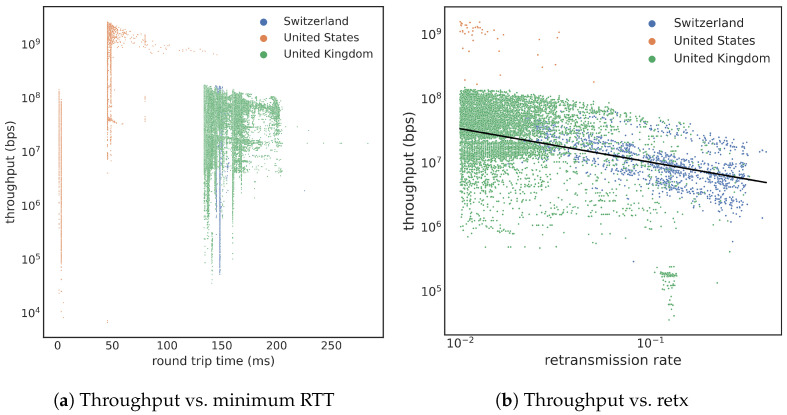
Throughput plotted against the minimum RTT and the retransmission rate (retx). Only shows three countries with the most transfers. Note that Figure 3b only shows transfers with retransmission rate greater than 0.01. Larger retransmission rates have noticeable impact on the throughput.

**Figure 4 sensors-23-05485-f004:**
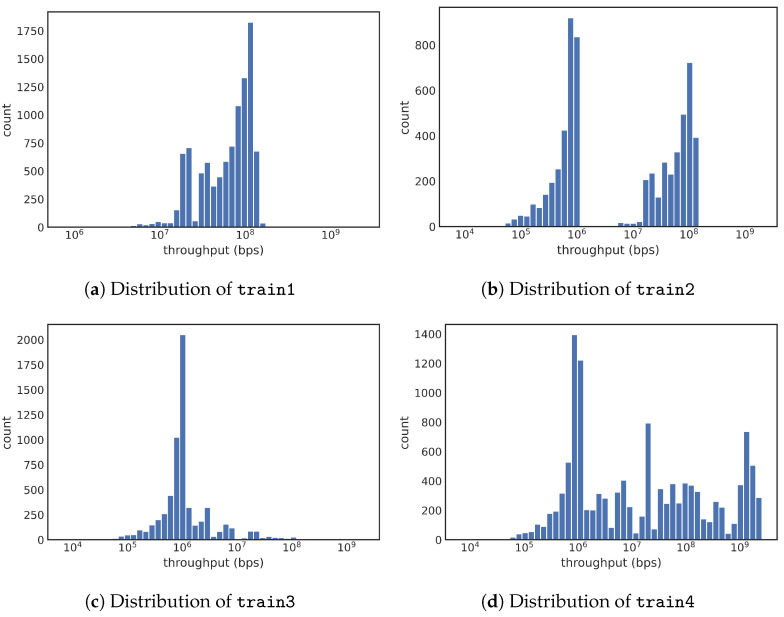
Distribution of throughput with different sampling strategies (Threshold = 1 Mbps): train1 is based on the uniform random sample, while the other three training sets are resulted from different stratified methods that keep the slow vs. normal classes in a balanced manner.

**Figure 5 sensors-23-05485-f005:**
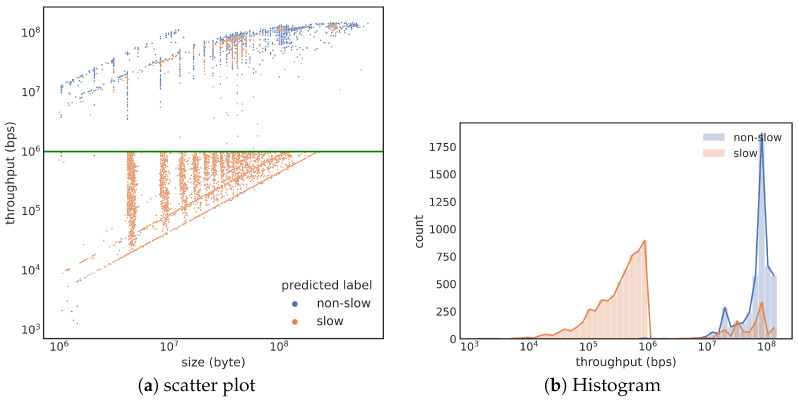
Prediction results of the train2-test2 pair. The misclassificiation appears primarily as counting some normal transfers as slow ones (orange color showing in the middle of blue).

**Figure 6 sensors-23-05485-f006:**
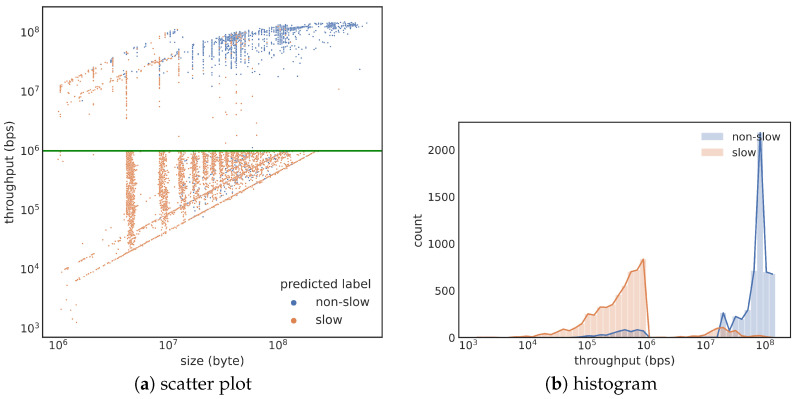
Prediction results of the train4-test2 pair. More misclassification cases are near the $10^{6}$ bps decision boundary than training with train2 shown in Figure 5.

**Figure 7 sensors-23-05485-f007:**
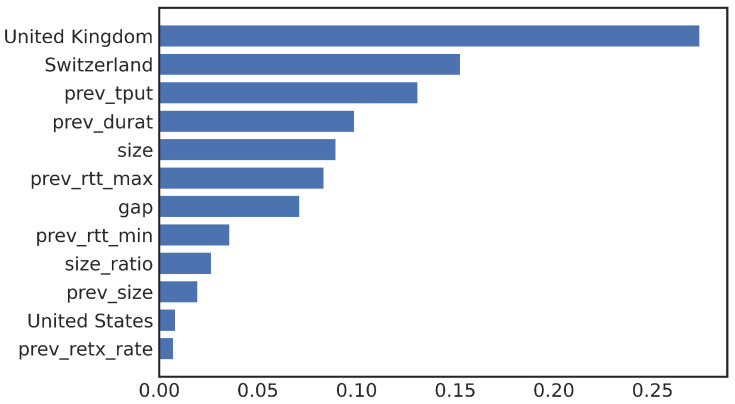
Top features from random forests.

**Figure 8 sensors-23-05485-f008:**
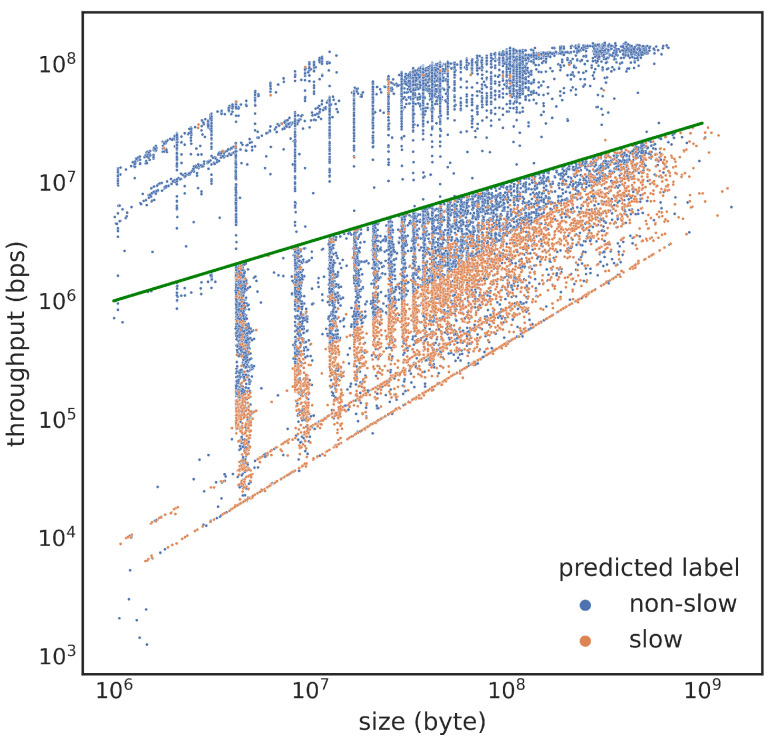
Scatter plot of throughput vs. transfer size: blue dots are classified as normal transfers, orange dots are classified as slow transfers, and the green line given by Equation (Equation 1) is the actual decision boundary used in training. Note that both axes are log-scaled.

**Table 1 sensors-23-05485-t001:** Number of file transfers to be studied: “large transfers” for transfers with size $>10^{6}$ bytes and “total transfers” for the number of network flows recorded by tstat. The  column of “ratio” contains the ratio between large transfers and total transfers.

DTN	Large Transfers	Total Transfers	Ratio
DTN01	737,498	172,657,723	0.43%
DTN02	39,917,258	163,310,125	24.43%
DTN03	137,272	86,034,525	0.16%
DTN04	101,119	72,204,065	0.14%
Total	40,893,147	494,206,438	8.3%

**Table 2 sensors-23-05485-t002:** Sampling strategies for organizing training and testing sets.

Feature	Collection	Description
train1	2021	uniform random sample from population (non-stratified)
train2	2021	keep the entire slow transfers and randomly sample the same number of normal transfers
train3	2021	keep the entire slow transfers and sample normal transfers progressively for each bin (so as to have more samples near the decision boundary)
train4	2021	keep the entire slow transfers and sample normal transfers with a fixed number for each bin
test1	2022	uniform random sample from population (non-stratified)
test2	2022	keep the entire slow transfers and randomly sample the same number of normal transfers
test3	2022	keep the entire slow transfers and sample normal transfers progressively for each bin (having more samples near the decision boundary)

**Table 3 sensors-23-05485-t003:** Features extracted from recently completed transfers to represent the network state for our prediction model.

Feature	Description
prev_tput	Latest throughput measured between the same src/dst networks (“a.b.c.0”)
prev_size	Latest transfer size (in bytes) between the same src/dst networks (“a.b.c.0”)
size_ratio	Ratio between the latest transfer size (prev_size) vs. current transfer size
prev_durat	Latest transfer duration (in msec) between the same src/dst networks (“a.b.c.0”)
prev_min_rtt	Latest minimum RTT between the same src/dst networks (“a.b.c.0”)
prev_rtt	Latest average RTT between the same src/dst networks (“a.b.c.0”)
prev_max_rtt	Latest maximum RTT between the same src/dst networks (“a.b.c.0”)
prev_retx_rate	Latest retransmission rate between the same src/dst networks (“a.b.c.0”)
time_gap	Time gap between latest vs. current transfers between the same src/dst networks (“a.b.c.0”)

**Table 4 sensors-23-05485-t004:** Testing results for all 12 train–test combinations ordered by F1 scores.

Train–Test Pair	F1 Score	Accuracy	Precision	Recall
train2-test2	0.926	0.925	0.921	0.931
train4-test2	0.907	0.907	0.905	0.908
train3-test2	0.885	0.888	0.914	0.857
train1-test2	0.875	0.888	0.988	0.785
train2-test3	0.709	0.590	0.550	0.996
train3-test3	0.682	0.533	0.518	0.998
train4-test3	0.682	0.533	0.518	0.998
train1-test1	0.566	0.999	0.778	0.444
train4-test1	0.415	0.998	0.303	0.659
train3-test1	0.340	0.998	0.285	0.421
train1-test3	0.252	0.559	0.845	0.148
train2-test1	0.234	0.995	0.145	0.603

**Table 5 sensors-23-05485-t005:** The features that appear most frequently in the top three best-performing decision trees trained from all four training datasets and tested with test2.

Feature	Number of Occurrence
prev_tput	12
country	10
size_ratio	9
prev_retx_rate	7
pre_rtt_min	6
prev_rtt_max	6

## Data Availability

This work makes use of summary information from operational logs of user accesses to NERSC. The current policies require such information to be kept within the center.

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
