# Peer review of "Leveraging History to Predict Infrequent Abnormal Transfers in Distributed Workflows"

_sensors, 2023, doi:10.3390/s23125485_

Round 1

Reviewer 1 Report

1. Plagiarism should be according to SOP. The single source should be less than 5%. 

2. Self-citation is not allowed. reference 12,14,21,22 and 29 detect with the same authors.

3. Author presents F1-score in this paper. but what is the accuracy and misrate of your proposed work?
4.  Add a comparison table in your work. 

5. Add a reference where you got datasets. 

6.  you must include the latest and relevant  literature review paper:

https://doi.org/10.1016/j.jksuci.2021.12.008

 https://doi.org/10.32604/cmc.2023.032617

Need to improve. 

Author Response

Thank you for the constructive feedback. We believe that your comments have been addressed in the revised manuscript. Please refer to the attached response document for details.

Reviewer 2 Report

The work focuses on predicting slow connections that create bottlenecks in distributed workflows. Authors devise several stratified sampling techniques to address the class imbalance challenge and study how they affect the machine learning approaches. Tests show that a relatively simple technique that under-samples the normal cases to balance the number of samples in two classes is very effective for model training. 

The topic fits the scope of the journal. 

The manuscript is well written, the structure of the paper is clear and the language is proper. 

The contributions are well delimited in the introduction section. However, I strongly suggest revisiting the up to date references in the topic covered by the paper. 

Additionally, authors should improve the discussion of  the related works in order to better improve the novelty of the work. 

Is there a related work which uses a similar method for feature engineering?

Results are well described supporting the objectives of the research.

The manuscript needs a revision in order to correct typos.

Minor editing of English language required

Author Response

(The authors gave the same response as above.)

Reviewer 3 Report

In this paper, several stratified sampling techniques to address the class imbalance challenge and study how they affect the machine learning approaches is proposed. Designed experiments showed the efficiency and effectiveness of the proposed methods. The investigated issue is interesting, and the paper shows the research depth and expertise. However, still minor revision is required as follows.

1. If the key novel contribution comes from the prediction, the prediction algorithms should be introduced more in the paper, and more details should be presented to show the novel design.

2. The dataset tstat includes the traffic statistics in what period, and does it contain the mobile data traffic.

3. For the evaluation, the dataset/algorithm choosing reasons, the detailed platform configurations and the discussion on other untested datasets should be introduced in the revised paper.

4. The technique part can be introduced deeply, in the current paper the description of this part is not enough.

5. Some references lack the necessary information (e.g., [2]), please provide all information according to the right template.

6. The length can be extended.

7. Please go through the paper carefully and double check whether the right template are used. Correct some typos and formatting issues (e.g., “restrainsmission-> “restransmission”? in Figure 3).

8. Make the References more comprehensive, besides this work, some other promising scenarios (e.g., Big data, other IoT systems) can be covered in this work. If the above related work can be discussed, it can strongly improve the research significance. For the improvement, the following papers can be considered to make the references more comprehensive.

Jin Wang, Changqing Zhao, Shiming He, Yu Gu, Osama Alfarraj, Ahed Abugabah, LogUAD: Log Unsupervised Anomaly Detection Based on Word2Vec, Computer Systems Science and Engineering, 2022, 41(3): 1207–1222

Lee, C. H., & Park, J. S., "An SDN-Based Packet Scheduling Scheme for Transmitting Emergency Data in Mobile Edge Computing Environments." Human-centric Computing and Information Sciences, vol. 11, article no. 28, 2021. https://doi.org/10.22967/HCIS.2021.11.028

Chubo Liu, Kenli Li, Keqin Li: A Game Approach to Multi-Servers Load Balancing with Load-Dependent Server Availability Consideration. IEEE Trans. Cloud Comput. 9(1): 1-13 (2021)

Khalil, A., Minallah, N., Ahmed, I., Ullah, K., Frnda, J., & Jan, N., "Robust mobile video transmission using DSTS-SP via three-stage iterative joint source-channel decoding." Human-centric Computing and Information Sciences, vol. 11, article no. 42, 2021. https://doi.org/10.22967/HCIS.2021.11.042

Babar, M., Khan, M. S., Habib, U., Shah, B., Ali, F., & Song, D., "Scalable Edge Computing for IoT and Multimedia Applications Using Machine Learning." Human-centric Computing and Information Sciences, vol. 11, article no. 41, 2021. https://doi.org/10.22967/HCIS.2021.11.041

The English writting of this paper is acceptable.

Author Response

(The authors gave the same response as above.)

Round 2

Reviewer 1 Report

Manuscript is fine.